# Influence of Traffic Parameters on the Spatial Distribution of Crashes on a Freeway to Increase Safety

Kamran Zandi [1,2], Ali Tavakoli Kashani [1,2,*] and Atsuyuki Okabe [3]

1   School of Civil Engineering, Iran University of Science and Technology, Tehran 16846-13114, Iran
2   Road Safety Research Center, Iran University of Science and Technology, Tehran 16846-13114, Iran
3   School of Global Studies and Collaboration, Aoyama Gakuin University, Tokyo 252-5258, Japan
*   Correspondence: alitavakoli@iust.ac.ir

**Abstract:** Significant research has been conducted in recent years to determine crash hotspots. This study focused on the effects of various traffic parameters, including average traffic speed and traffic volume, on the spatial distributions of freeway crashes. Specifically, this study analyzed the spatial distributions of crashes on the Qazvin–Abyek freeway in Iran using four-year crash records. Spatial crash clustering analysis was performed to identify hotspots and high cluster segments using global Moran's *I*, local Moran's *I*, and Getis-Ord *Gi\**. The global Moran's *I* indicated that clusters were formed under the low range of hourly traffic volume (less than 1107 veh/h) and the high range of traffic speed (more than 97 km/h), which increased the number of heavy vehicle crashes in the early morning (time 03–06) around the 52 km segment. The results obtained from kernel density estimation (KDE), local Moran's *I*, and Getis-Ord *Gi\** revealed similar crash hotspots. The results further showed different spatial distributions of crashes for different traffic hourly volumes, traffic speed, and crash times, and there was hotspot migration by applying different traffic conditions. These findings can be used to identify high-risk crash conditions for traffic managers and help them to make the best decisions to enhance road safety.

**Keywords:** spatial distribution of crashes (SDC); spatiotemporal distribution of crashes (STDC); freeway crashes; crash hotspot





## 1. Introduction

Road crashes, which cause high rates of casualties, injuries, and financial costs, have been significant problems for the governments of developing countries, including Iran. In 2020, there were about 15,400 crash fatalities in Iran and, as reported by the Iranian Forensics Organization, around two-thirds of them occurred on suburban roads—especially on freeways.

In the last decade, many studies have been conducted on the spatial distribution of crashes (SDC) and the temporal distribution of crashes (TDC) to investigate the causes of crashes and identify crash hotspots [1,2]. Their findings prove that there is no uniform SDC and TDC (e.g., [3,4]). These results mean that there is a non-uniform distribution of crashes on different parts of roads for various reasons. Moreover, the distribution of crashes is not the same at different times, such as months of the year, days of the week, hours of the day, etc. Many studies have been conducted on the spatiotemporal distribution of crashes (STDC) [3–7]. For instance, Li and Zhu [2] and Plug and Xia [3] suggest an interaction between the time and location of crashes based on STDC. Moreover, the spatiotemporal analysis indicates the variations in the spatial distribution of relative crash risks at different periods, such as hours of a day, days of a week, months of a year, and different years [8,9].

Many studies explore the influence of certain parameters on road traffic safety, including behavioral factors and driver characteristics [10–12]. For instance, Besharati and Tavakoli Kashani [13] argue that intercity bus drivers' behavioral characteristics significantly affect crash risks. The effects of behavioral factors and drivers' characteristics on

SDC and temporal distribution of crashes (TDC) have also been identified [14,15]. Moreover, Toran Pour and Moridpour [16] considered age and gender as parameters that affect pedestrians' SDC and TDC. Toran Pour and Moridpour [16] also suggested different spatial distribution patterns for different human characteristics, whether for drivers or pedestrians. Anvari and Kashani [17] proved that passenger factors and the rider's age affect rear-end motorcycle collisions, and that age and gender have a significant influence on children's pedestrian behavior in conflict zones of urban intersections [18]. The effect of weather conditions on the SDC has also been confirmed [19].

Furthermore, several studies have focused on investigating the effects of traffic parameters such as speed and traffic volume on crash frequency and severity (e.g., [20–22]). According to Quddus [23], a 1% increase in average annual daily traffic (AADT) is associated with a 0.5% increase in the rates of killed and seriously injured (KSI) casualties. An increase in average traffic speed causes an increase in crash frequency. For example, a 10% increase in the average speed leads to a 30% increase in fatalities and crash severity [24].

Since the present study focuses on SDC and STDC, another attempt led to investigating the effects of traffic parameters such as hourly traffic volume and average traffic speed on SDC. Kashani and Zandi [25] found that there were entirely different TDC patterns for different hourly traffic volumes and average traffic speeds. In other words, these parameters had a significant effect on TDC. In addition, Plug and Xia [3,4] highlighted that speed is a factor that can affect TDC. Salem and Genaidy [26] also found that increasing the speed variance has an impact on SDC and increases rear-end crashes in the work zone. However, to the best of the authors' knowledge, only a few researchers have focused on the impacts of some other traffic parameters. If the effects of traffic parameters such as average hourly traffic volume and average point speed on the SDC are proven, they can be used to identify accident hotspots in different traffic conditions. Hence, they can help traffic managers to identify conditions that pose a high crash risk. It should be noted that in the analyses, the point traffic information at the crash location is employed. However, since at the last stage of the procedure this information is related to a specific origin (and not separate road segments), it can be regarded as spatial information. This study further tried to identify the factors leading to crashes at a particular location and time. Additionally, it aimed at finding the spatiotemporal hotspots.

There are several methods for spatial analysis, including kernel density estimation (KDE), Getis-Ord *Gi\**, k-means clustering, and the nearest neighbor method [27]. As there was point pattern crash analysis, the KDE method was used, because it is a suitable method for identifying hotspots [27,28]. Meanwhile, KDE is the most common and well-established method and is one of the most widely used analysis methods for density estimation [1,3].

Moran's *I*, Geary's C, and Getis-Ord *Gi\** can be employed to find spatiotemporal autocorrelation by providing a single value of spatial autocorrelation, but they do not present the locations of clusters [16,29,30]. Despite suggesting the Bayesian approach for the spatiotemporal patterns of relative crash risks [2], several studies (e.g., Plug et al., 2011 [3]; Vemulapalli et al., 2016 [27]) have employed the co-map method for spatiotemporal analysis, and the present study was no exception. Co-map is an extension of co-plot that is a visualization tool [4]. Co-plot uses point plots and a spatial distribution pattern map, such as the KDE, to reveal the effect(s) of changing two variables [28]. Co-map can be used to investigate the temporal and spatial patterns of an event, including crime, vehicle crashes, and disease. Soltani and Askari [31] highlighted that the basis of the spatiotemporal analysis is dividing the overall parameter into subsets, and these subsets are then plotted to examine the differences among each subset. The significant relationship between SDC patterns and different conditions could help network administrators to identify specific situations to adopt proper decisions to reduce the risks of crashes.

## 2. Methodology

The methodology of the present study is shown as a flowchart in Figure 1. It is worth mentioning that steps 3 and 4 have been previously performed by the authors and can

also be followed in the work of Kashani and Zandi [25]; the rest of the steps are reported in this study. The other steps shown in Figure 1 were the main focus of this research, as follows: Step 1 is obtaining the crash and traffic datasets, and step 2 is the merging of these two data series. More explanations of steps 1 and 2 can be found in Section 3 (Case Study and Data). Step 5 is about spatial analysis, while step 6 is about testing significance and its methods; both steps are described in Section 2.1 (Spatial Analysis) and Section 2.2 (High-Frequency Crash Location). Step 7 investigates the spatiotemporal analysis, which explains the correlation of time and location of the crashes, as described in Section 1 (Introduction). In the rest of this section, the different parts of the methodology are explained.

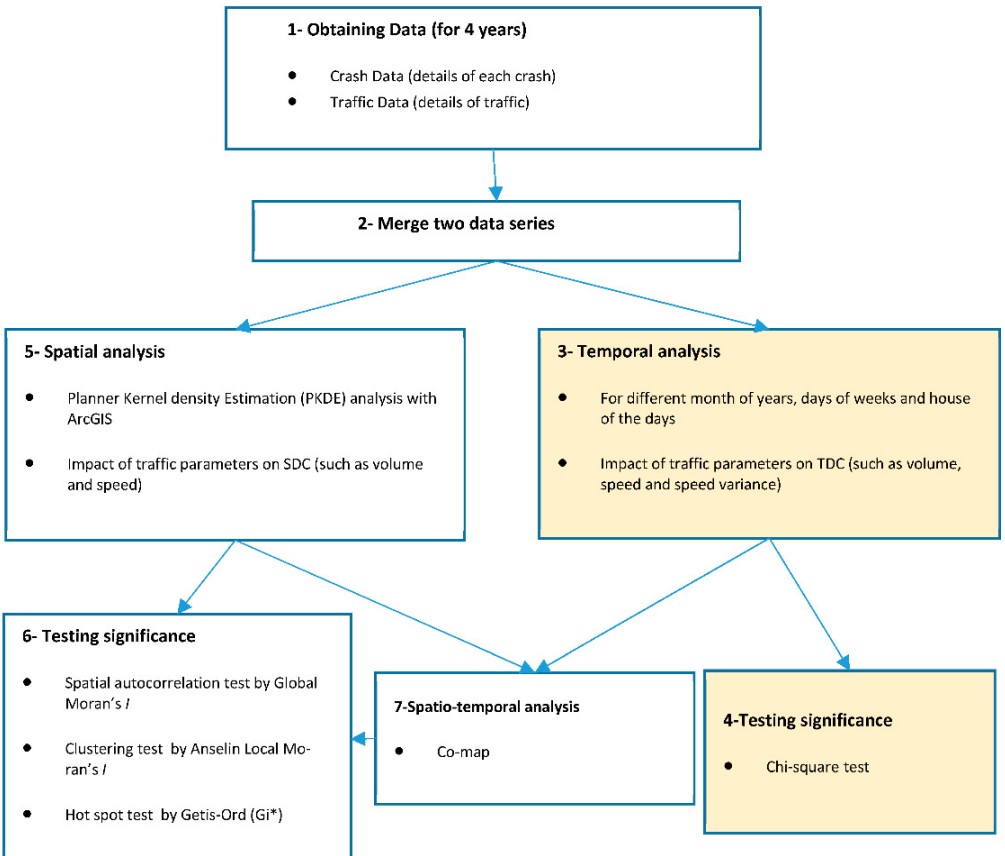

**Figure 1.** Flowchart of the methodology.

### 2.1. Spatial Analysis

For spatial analysis, since KDE analysis has been used in many previous analyses, it was also employed in this research. KDE is a distance-based method generating a smooth and continuous surface map of the risk target [32]. KDE calculates a density surface within a circular distance centered at each point across a study area for identifying hazardous road situations and hotspot point events. The idea is that a crash may occurs not only at a point but with a pattern around that point, [27,29,33,34]. "Crashes on a plane within the kernels are weighted based on their Euclidean distance from the kernel center, and the resulting density value is assigned to that center. The distance is weighted according to a kernel function" [29].

For spatial analysis, both kinds of KDE—i.e., planar (2D) kernel density estimation (PKDE) and network kernel density estimation (NKDE)—are useful. PKDE reveals the density on the surface; on the other hand, NKDE shows the density along a network [35]. "NKDE is an extension of PKDE that calculates the density of point type events on a linear unit (in a network) rather than on a 2-D homogeneous area unit, as was the case of PKDE" [34].

PKDE was analyzed using ESRI ArcMap 10.4.1 software, while NKDE was estimated using SANET software [36]. Since both methods can help identify clusters, PKDE was employed more in the present study. PKDE analysis can be written as shown in Equation (1):

$$\lambda(s) = \sum_{i=1}^{n} \frac{1}{\pi r^2} k\left(\frac{d_{is}}{r}\right), \tag{1}$$

where $\lambda(s)$ is the density in the location, $s$ and $r$ represent the radius (bandwidth) of KDE, and $k$ denotes the weight of point $i$ at the distance $d_{is}$ of location $s$ [34]. On the other hand, $k$ denotes the population value for each feature; in this research, the $k$ parameter was defined to the software as the crash number for each point of the path.

Moreover, "the network KDE is very similar to the planar KDE, except that the normalizing constant is $r$ instead of $\pi r^2$" [28].

A significant point in the KDE analysis is the consideration of the appropriate horizontal bandwidth value that can change the quality of the kernel estimate. To identify the optimal bandwidth, the distance increment tests for the global Moran's $I$ were run for different conditions and analyzed using ESRI ArcMap 10.4.1 software.

Another objective of the present study was to examine the spatial autocorrelation of traffic crashes. Strong spatial autocorrelation indicates a spatial relationship between crashes; otherwise, the crashes will occur randomly, and there is no relationship between them. As the global Moran's $I$ method has proven helpful for estimating the spatial autocorrelation of crashes [29,31], it was employed in this study. The global Moran's $I$ for autocorrelation can be written as shown in Equation (2):

$$I = \frac{n}{S_0} \frac{\sum_{i=1}^{n} \sum_{j=1}^{n} w_{i,j} z_i z_j}{\sum_{i=1}^{n} z_i^2}, \tag{2}$$

where $z_i$ is the deviation of an attribute for feature $i$ from its mean ($x_i$- $\overline{X}$) (Equation (3)), $w_{i,j}$ denotes the spatial weight between features $i$ and $j$, $n$ is the total number of features, and $S_0$ is the aggregate of all spatial weights (Equation (4)):

$$z_i = \frac{I - E[I]}{\sqrt{V[I]}}, \tag{3}$$

$$S_0 = \sum_{i=1}^{n} \sum_{j=1}^{n} w_{i,j}, \tag{4}$$

where $E[I]$ is the expected value of Moran's $I$ under the null hypothesis of no spatial autocorrelation, and $V[I]$ is the variance I (Equations (5) and (6)):

$$E[I] = -1/(n-1), \tag{5}$$

$$V[I] = E\left[I^2\right] - E[I]^2. \tag{6}$$

The spatial autocorrelation (global Moran's $I$) was also estimated using ESRI ArcMap 10.4.1 software. This software measures the spatial autocorrelation based on both locations and value features to check whether the pattern expressed is clustered, dispersed, or random. It also calculates Moran's $I$ index values and $p$-values to evaluate the significance of that index. The $w_{i,j}$ could be the crash frequency that was defined for the software as the input field (i.e., the total number of crashes or the number of crashes in the specific condition) for each point of the path.

### 2.2. High-Frequency Crash Locations

The spatial autocorrelation tools can identify high-frequency crash locations. They help to simultaneously measure spatial autocorrelation based on feature locations and

feature values. Given a set of features and their associated attribute values, they evaluate whether a given pattern is clustered, dispersed, or random. Strong autocorrelation occurs via the global Moran's *I* when the values of geographically close cells are similar. Although the global Moran's *I* is a method to define spatial autocorrelation for a whole path, it cannot determine the location of the clusters locally and only provides a quick view of the condition of the spatial dependencies.

Many analysts are interested in knowing clusters' locations and displaying them on a map, even if there is no autocorrelation. Even if there is no global spatial autocorrelation, there may still be spatial autocorrelation on some parts of the way [37]. Various cluster mapping tools have been employed for cluster analysis. With the help of these tools, places with statistically significant hot, cold, and unpleasant spatial spots can be determined.

Various cluster mapping tools can help to determine places with statistically significant hot, cold, and unpleasant spatial spots. Local indicators of spatial association (LISA) provide a measure for each spatial unit's association and help identify the type and location of clusters [7,38,39]. However, KDE is a tool for identifying clusters. Other methods—such as KDE+, Getis-Ord *Gi**, and Anselin local Moran's *I*—are also used to determine hotspots. The KDE+ method, as stated by Bíl and Andrášik [40], is based on the principles of KDE that can be employed to find significant clusters and hotspot rankings.

The Getis-Ord *Gi** is a widespread and valuable method to identify the spatial clusters of high value (hotspots) and of low value (cold spots) [1]. The Getis-Ord *Gi** equation can be written as shown in Equation (7) [28]:

$$G_i^* = \frac{\sum_{j=1}^{n} w_{i,j}(D) X_j - \overline{X} \sum_{j=1}^{n} w_{i,j}}{S\sqrt{\frac{\left[n \sum_{j=1}^{n} w_{i,j}^2 - \left(\sum_{j=1}^{n} w_{i,j}\right)^2\right]}{n-1}}}, \tag{7}$$

where *Gi** is the Getis-Ord *Gi** value for feature *i*, $w_{i,j}(D)$ denotes the weight between features *i* and *j* (i.e., the input field of the software that could be defined as the crash frequency for the desired condition), *D* is the distance between features *i* and *j*, $X_j$ shows the frequency at location *j*, n reflects the number of all features, $\overline{X}$ represents the average crash frequency, and *S* is the standard deviation of $X_j$.

The *Gi** statistic is evaluated with a z-score. A higher positive z-score shows a high value of clustering, while a smaller negative z-score indicates a low value of clustering or a cold spot.

"To investigate the spatial variation as well as the spatial associations, it is possible to calculate local versions of Moran's *I*" [30]. The local Moran's *I* is another method for detecting high and low clusters, as well as high and low outliers. The local Moran's *I* statistics of spatial association are given in Equation (8):

$$I_i = \frac{x_i - \overline{X}}{S_i^2} \sum_{\substack{j=1 \\ j \neq i}}^{n} w_{i,j}(x_i - \overline{X}), \tag{8}$$

where $x_i$ is an attribute for feature *i*, $\overline{X}$ represents the mean of the corresponding attribute, $S_i$ is the standard deviation, *n* is the total number of features, and $w_{i,j}$ denotes the spatial weight between features *i* and *j* (i.e., the input field of the software that could be defined as the crash frequency for the desired condition). If the value of *I* is positive, it suggests that the feature is surrounded by the same feature; thus, the feature is a part of that cluster. At the same time, the negative value of *I* indicates that the feature is surrounded by a feature that is not similar. This type of feature is called an outlier. The Getis-Ord *Gi** and the local Moran's *I* index were calculated only within the standard rating framework, where the *p*-value can be interpreted and analyzed.

### 3. Case Study and Data

This research focused on a 56 km freeway between Qazvin and Abyek in Qazvin Province. This freeway connects Tehran to four provinces directly and to five other provinces indirectly. On the other hand, this freeway is also important for international transit between Iran and its northern and northwestern neighbors (Figure 2). This is a freeway in Iran with a very high crash rate (nearly 330 fatal crashes annually) and annual traffic volume (about 31 million vehicles per year). The details of the number of crashes over the past nine years are shown in Table 1.

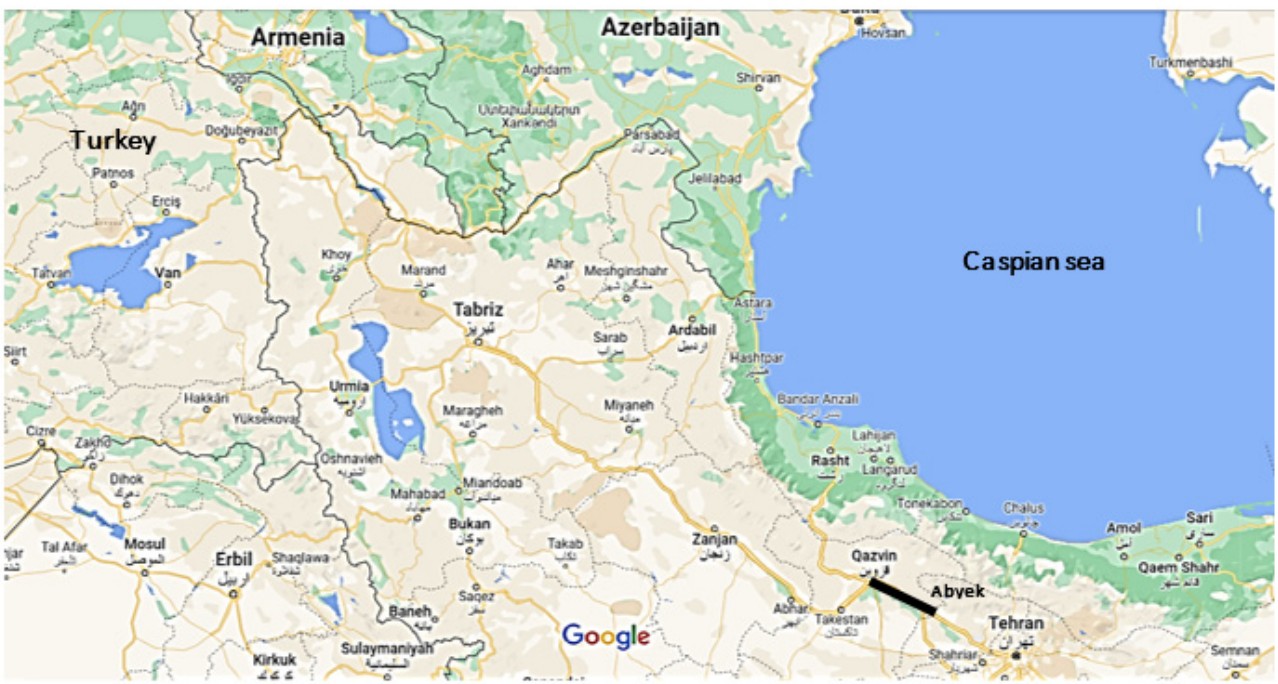

**Figure 2.** Qazvin–Abyek freeway location. Reference: Google Maps.

**Table 1.** Summary of the numbers of deaths and injuries between 2011 and 2020.

| Direction | Crashes | No. of Deaths | No. of Injuries |
| --- | --- | --- | --- |
| Qazvin to Abyek | 1550 | 204 | 2165 |
| Abyek to Qazvin | 1708 | 193 | 1993 |
| Total | 3258 | 397 | 4158 |

There were two datasets on this freeway's crashes and traffic conditions. The first dataset, obtained from the traffic police of Qazvin Province, detailed the last four years' crash data, including the location (the crash segment), time, date, weather conditions (i.e., normal or abnormal conditions), details of the vehicles involved in the crash, drivers' and accident victims' characteristics (including gender and age), and types of crashes. The second dataset contained hourly traffic volumes and average traffic speeds for all types of vehicles (including cars, trucks, and trailers) on all parts of the freeway. The Iran Road Maintenance and Transportation Organization obtained these datasets via the mounted cameras and loop detectors along the route. More information about each crash was obtained by merging these two datasets. This means that some traffic data on the times and locations of crashes were further added to the crash data; thus, each record had more information about the conditions of the crashes. This study focused on crash fatalities and injuries.

## 4. Results and Discussion

### 4.1. SDC Analysis

The PKDE method was run for both directions (Qazvin to Abyek and Abyek to Qazvin), and the results are shown in Figures 3 and 4. The local Moran's *I* and Getis-Ord *Gi\** methods were employed to locally identify crash clusters and hotspots. The distance increment test was run for different conditions to determine the optimal bandwidth for the global Moran's *I*. Both very small and very large bandwidths could be useless. Different bandwidths are suggested for urban areas and freeways [29,41]. The results of the distance increment test can be seen in Table 2, which also contains the Moran's *I* index, z-score, and *p*-value with 95% confidence. The results indicated that the optimal bandwidth with the highest z-score was a distance of about 700 m for Qazvin to Abyek and 800 m for Abyek to Qazvin. Since the length of the investigated crash segments was 1000 m, the 1000 m bandwidth was tested. The results indicated an acceptable *p*-value with 95% confidence. Thus, similar to the findings of other studies, such as the work of Mohaymany and Shahri [29], the present study's bandwidth was set to 1000 m.

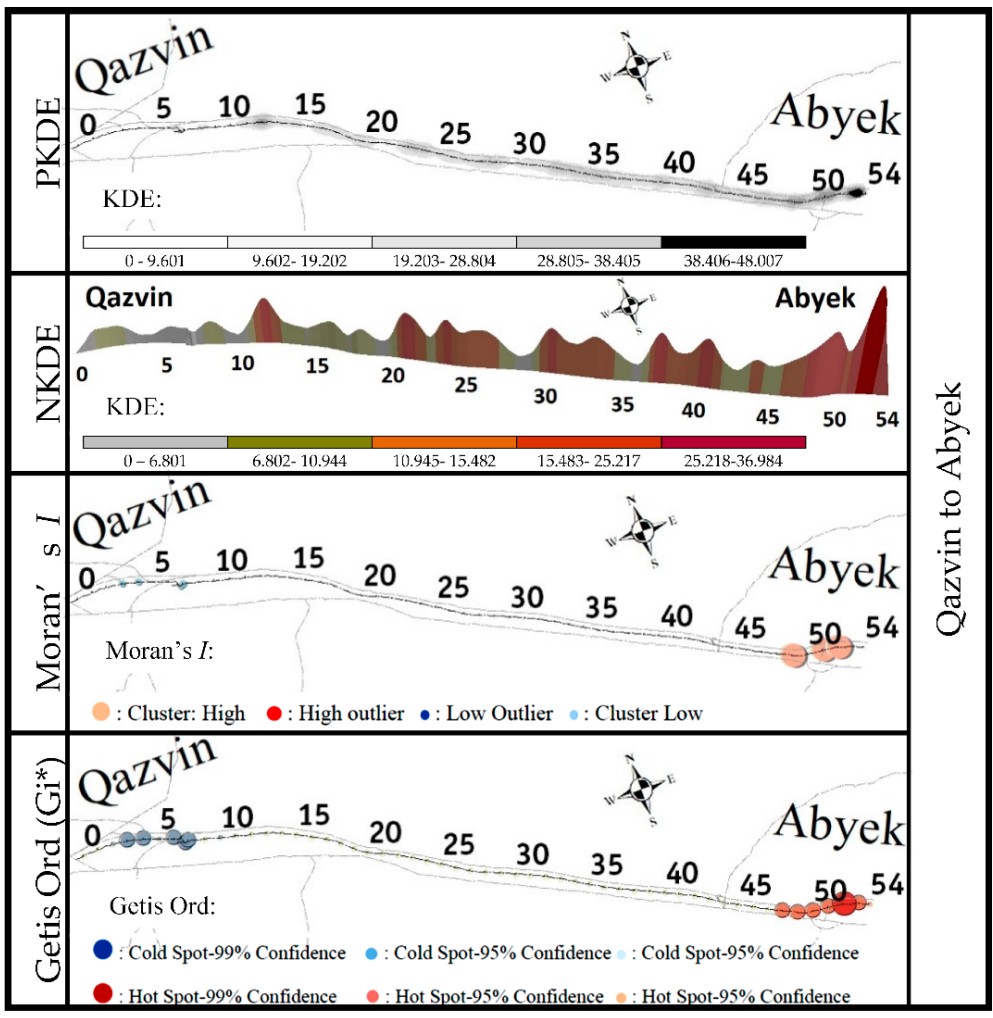

**Figure 3.** Different spatial analysis outputs of Qazvin-to-Abyek freeway crashes. Note: Segment numbers on the paths are the distance in km.

For Qazvin to Abyek, as illustrated in Figure 3, the hotspot segments identified by the Getis-Ord *Gi\** method with at least 95% confidence are 48, 49, 50, 51, 52, and 53, while the high clusters identified by the Moran's *I* method are 49, 51, and 52. For Abyek to Qazvin, the hotspot segments obtained from the employment of the Getis Ord *Gi\** method with at least 95% confidence are 50, 51, and 52, whereas the high clusters determined via the

Moran's *I* method are 51 and 52 (Figure 4). In segment 6 of the Abyek–Qazvin freeway, some points seemed to be hotspots in PKDE, while not being high clusters or hotspots in the Getis-Ord *Gi\** and Moran's *I*. Thus, it can be concluded that these methods did not necessarily lead to the same results.

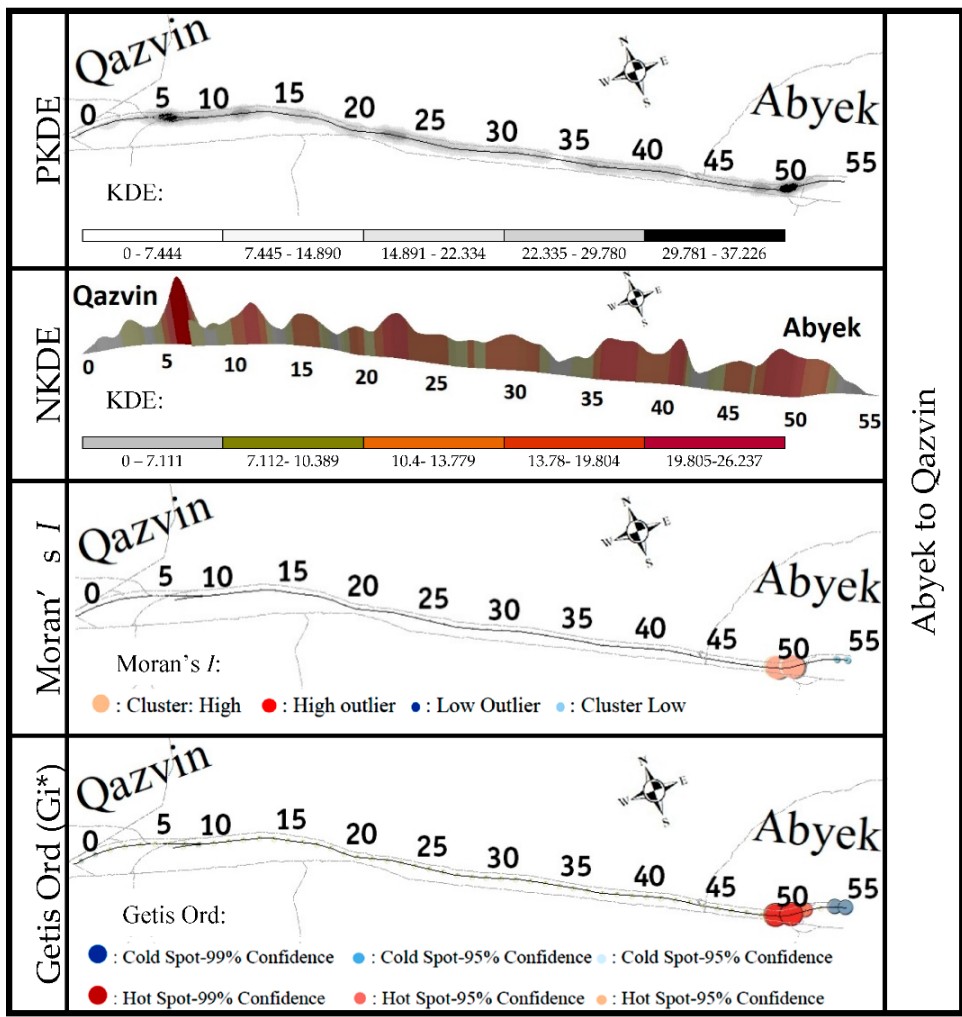

**Figure 4.** Different spatial analysis outputs of Abyek-to-Qazvin freeway crashes. Note: Segment numbers on the paths are the distance in km.

**Table 2.** Incremental bandwidth test for global Moran's I.

| | Direction | | | Direction | | |
|---|---|---|---|---|---|---|
| | **Qazvin to Abyek** | | | **Abyek to Qazvin** | | |
| Distance (m) | Moran's Index | z-Score | *p*-Value | Moran's Index | z-Score | *p*-Value |
| 500 | 0.280477 | 14.45204 | 0.00 | 0.262819 | 13.56918 | 0.00 |
| 600 | 0.253837 | 14.58366 | 0.00 | 0.235518 | 13.48739 | 0.00 |
| 700 | 0.230517 | 14.72402 | 0.00 | 0.204491 | 13.1025 | 0.00 |
| 800 | 0.194255 | 13.36734 | 0.00 | 0.197075 | 13.54769 | 0.00 |
| 900 | 0.167901 | 12.22441 | 0.00 | 0.168325 | 12.33501 | 0.00 |
| 1000 | 0.152191 | 11.81901 | 0.00 | 0.145903 | 11.38119 | 0.00 |
| 1100 | 0.163536 | 13.45219 | 0.00 | 0.12565 | 10.28813 | 0.00 |
| 1200 | 0.147651 | 12.63698 | 0.00 | 0.105662 | 9.085491 | 0.00 |

### 4.2. STDC Analysis

The spatiotemporal analysis of crashes was carried out by the co-map method. This method helps to compare temporal crash distribution for different conditions. The temporal part was performed with the spider plot method, consisting of two parts: one for all crashes, and the other for Friday as the weekend and Tuesday as a weekday. The data illustrated on the left side of Figure 5 indicate that the hot times for all crashes are 15, 14, and 18, while they are 08 for Tuesday and 13, 14, and 20 for Friday [25]. Of course, these times may be different for urban crashes. As an example, for urban crashes in London, the high crash times were 8:00–11:59 and 16:00–19:59 for all days of the week [42]. Figure 5 also demonstrates different TDC patterns for different conditions.

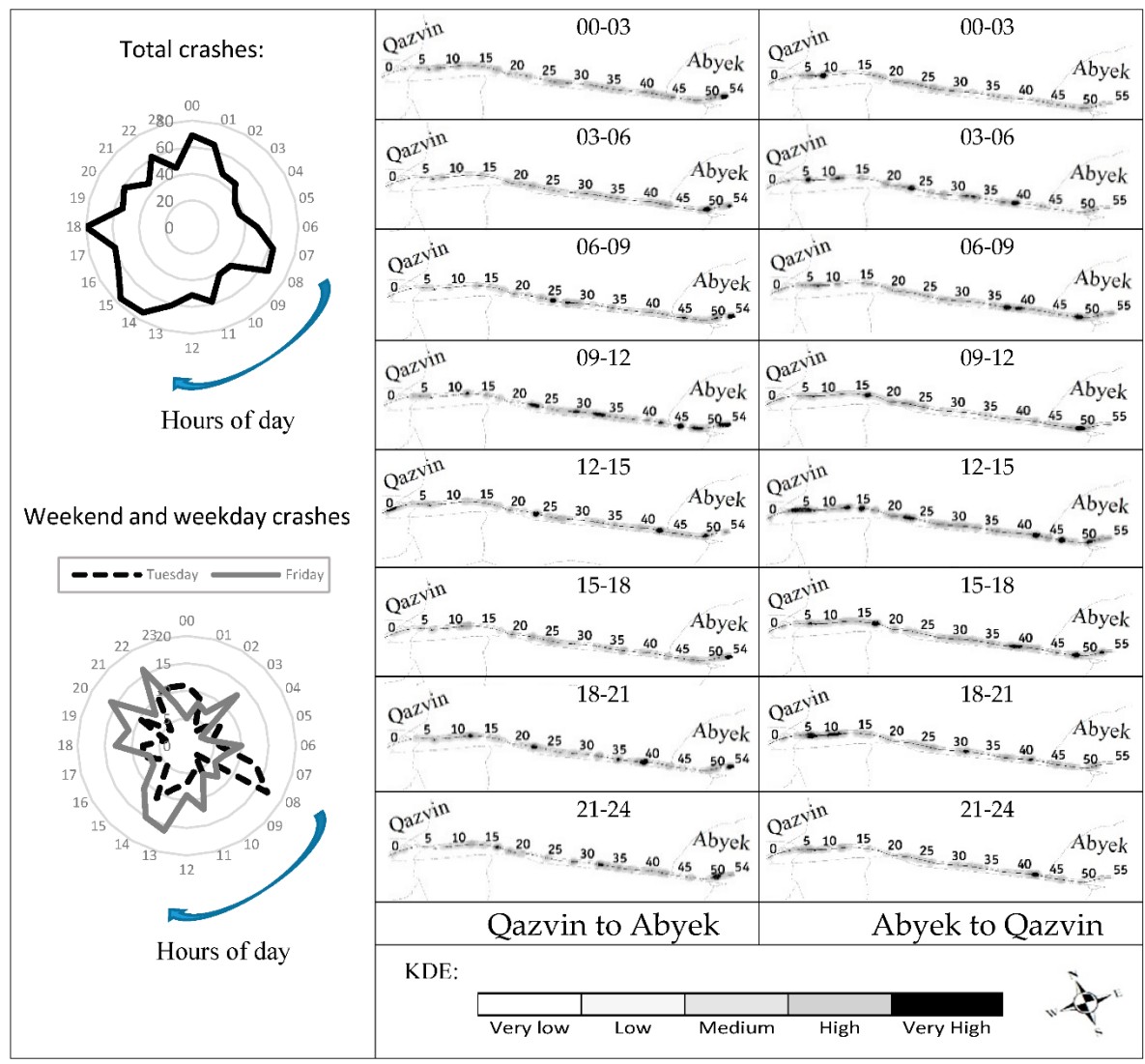

**Figure 5.** Co-map of spatiotemporal analysis of crashes. Note: Segment numbers on the paths are the distance in km.

To estimate the autocorrelation between the locations and times of crashes—i.e., spatiotemporal analysis of crashes—the hours of the days were initially divided into eight equal parts. The PKDE analysis was then run for the spatial part of the analysis and for each time and direction. The right side of Figure 5 shows different hotspots for different parts of the day, i.e., morning, night, midnight, and noon. The local Moran's *I* and Getis-Ord *Gi\** methods determined high clusters and hotspots (Figure 6). Comparing the local Moran's *I*, Getis-Ord *Gi\**, and PKDE methods revealed many similarities. Moreover, the

spatiotemporal autocorrelation analysis of the crashes showed different hotspots and high clusters for different times and locations.

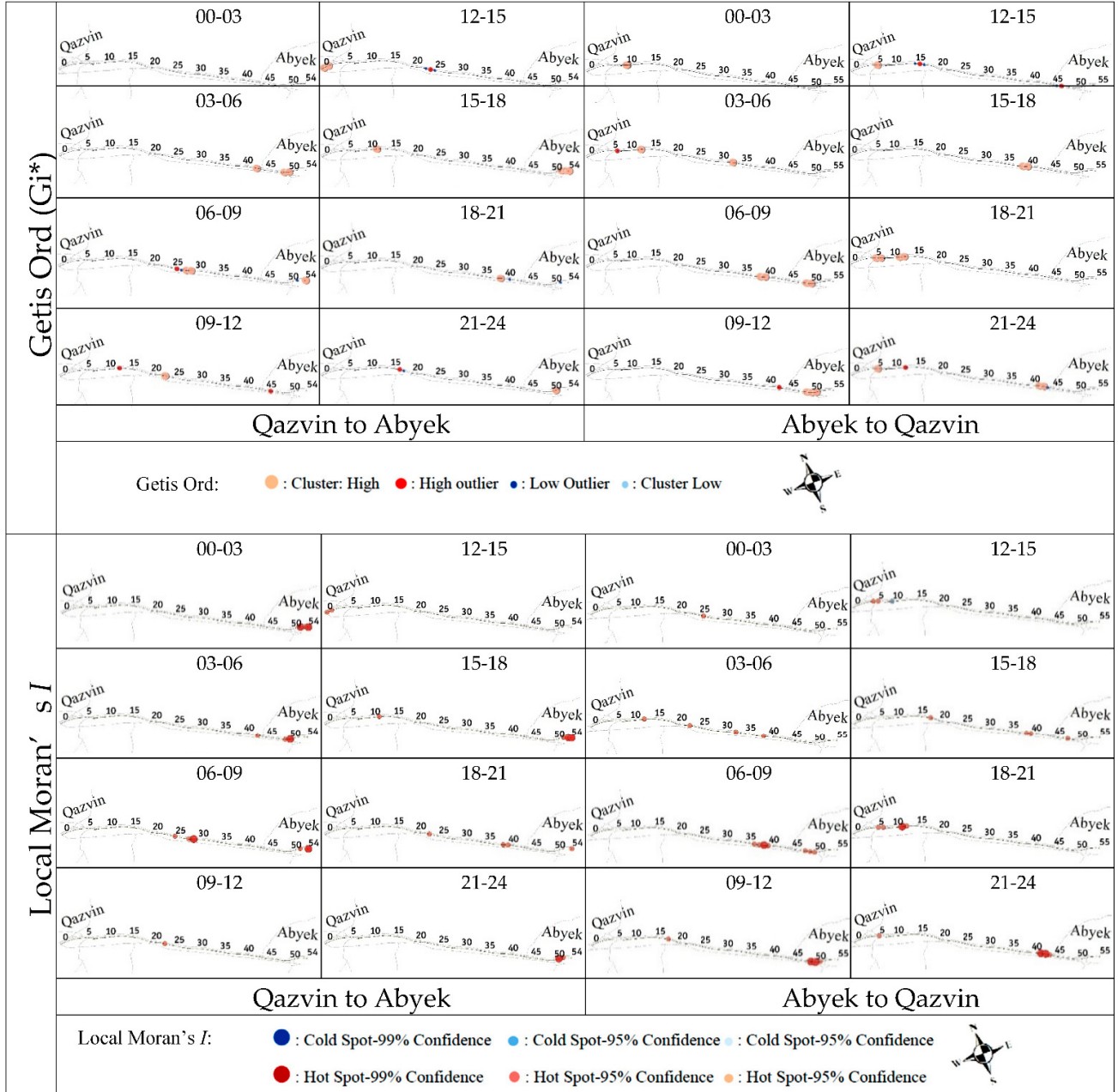

**Figure 6.** Local Moran's *I* and Getis-Ord *Gi*\* for STDC. Note: Segment numbers on the paths are the distance in km.

### 4.3. Influence of Traffic Parameters on the SDC

The main objective of this part was to investigate whether there were any significant differences between hotspots and high clusters for different hourly traffic volumes and traffic speeds. To examine the effect of hourly traffic volume on the SDC, the entire crash data were divided into three parts—low, medium, and high hourly traffic volume—so that the numbers of crashes in them were almost the same. These values and the co-map results can be seen in Figure 7, including PKDE, local Moran's *I*, and Getis-Ord *Gi*\* for both directions. The results show significant SDC differences among different volume ranges, proving the effect of hourly traffic volume on the SDC.

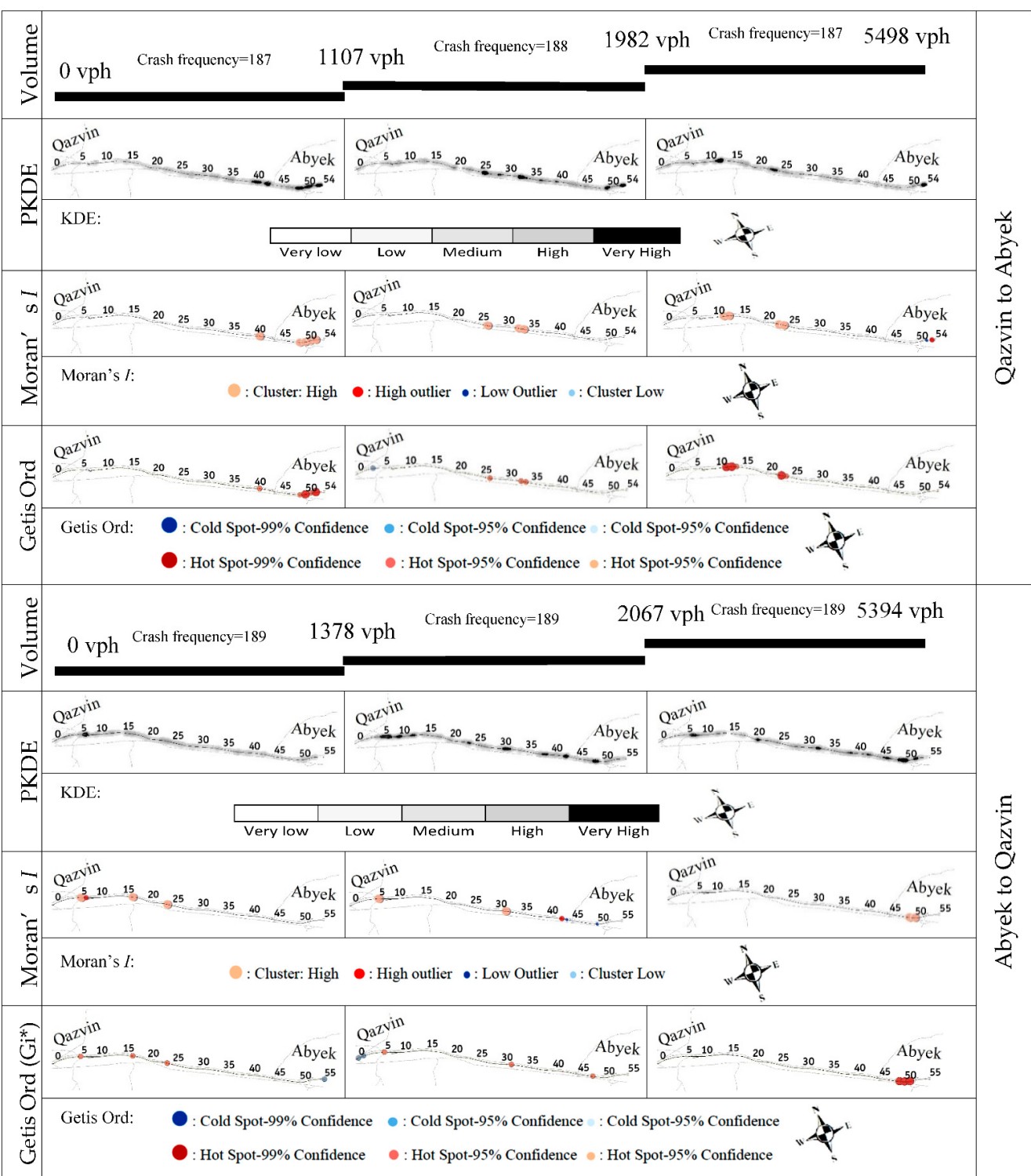

**Figure 7.** Impact of hourly traffic volume on the spatial distribution of crashes (SDC). Note: Segment numbers on the paths are the distance in km.

Similarly, the effect of average speed on the SDC was estimated. By investigating the crash frequency and mean traffic speed, it was found that the highest crash frequency occurs at nearly 95 km/h speed. In other words, the speed of 95 km/h is a marginal value at which the rates of crashes before and after it are completely different. Then, the crash data were divided into two classes: low speed, and high speed. To retain the same terms of comparison, an average speed divide point was selected to give an almost equal crash

frequency (97 km/h average speed). Thus, there were two crash data classes: speeds lower and higher than 97 km/h. As illustrated in Figure 8, the results were almost identical to those of the previous step. Therefore, there were different shapes of the SDC for different traffic speeds.

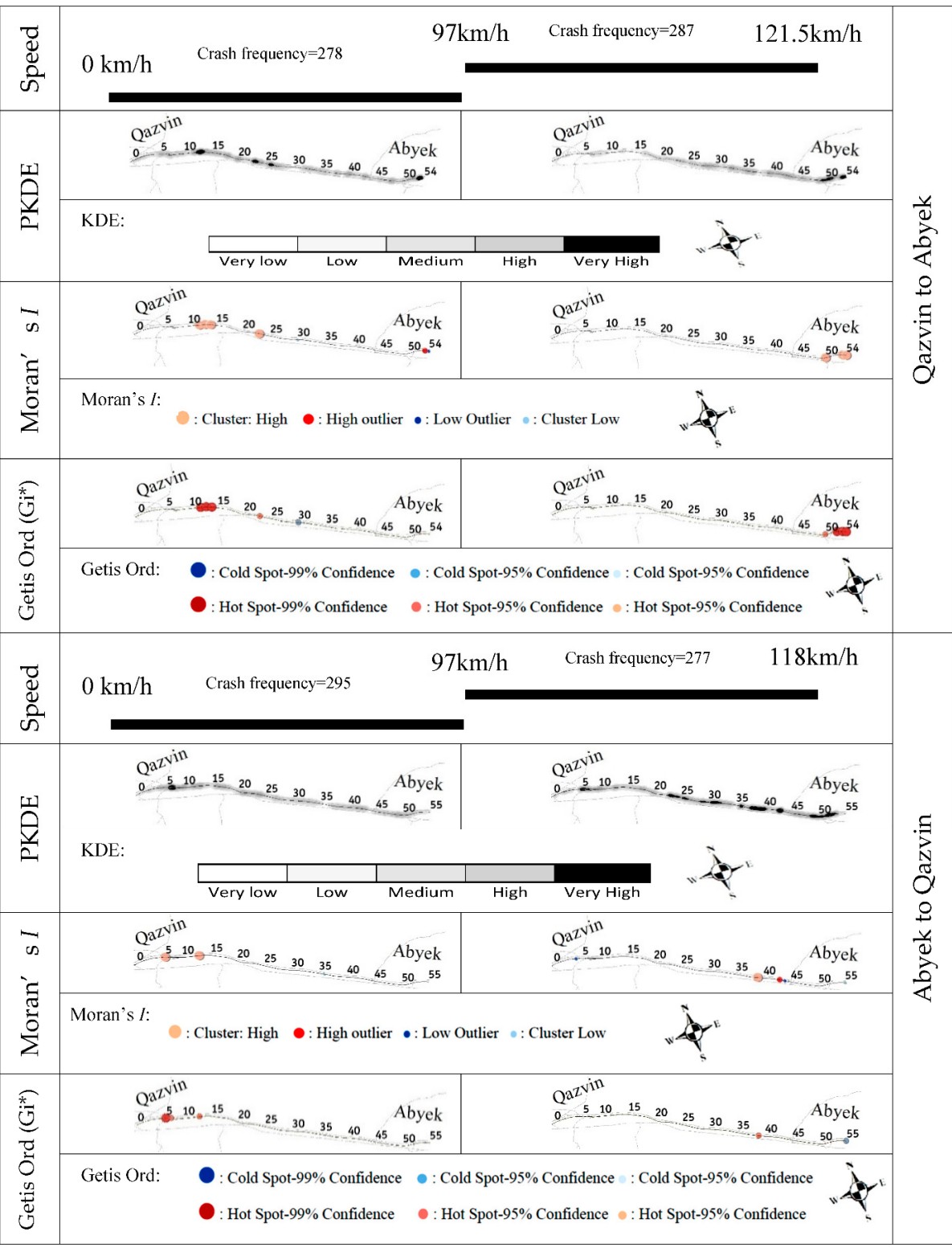

**Figure 8.** Impact of average traffic speed on the SDC. Note: Segment numbers on the paths are the distance in km.

For more investigating details, the impact of average hourly volume on SDC is provided in Figure 9, which focuses on the local Moran's *I* hotspot migration on the Abyek–Qazvin freeway for two different conditions: low and high average traffic volume. The results show the hotspot migration from sections 5, 17, and 24 to sections 49 and 50. More information on the impact of average traffic speed on the Qazvin–Abyek freeway's SDC is provided in Figure 10. There is hot spot migration from sections 12, 13, 14, and 23 to sections 50, 53, and 54. This proves a significant impact of both traffic volume and average speed on the SDC.

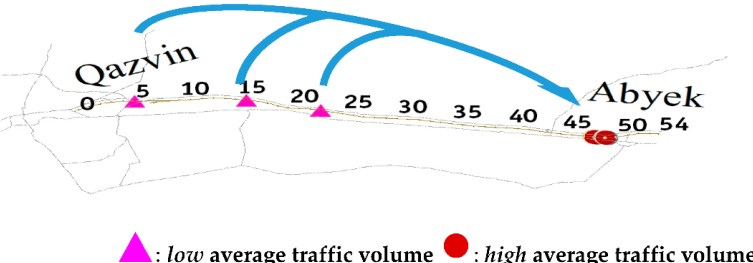

▲: *low* **average traffic volume**　●: *high* **average traffic volume**

**Figure 9.** Hotspot migration from low to high average hourly traffic volume from Abyek to Qazvin. Note: Segment numbers on the paths are the distance in km.

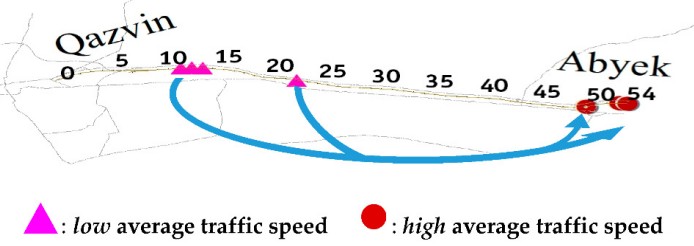

▲: *low* **average traffic speed**　●: *high* **average traffic speed**

**Figure 10.** Hotspot migration from low to high average traffic speed from Qazvin to Abyek. Note: Segment numbers on the paths are the distance in km.

Figures 9 and 10 show two examples of crash hotspot migration in different traffic conditions, which can also be extended to other conditions. Some further crash hotspot migrations can be found in Appendix A (Tables A1 and A2).

### 4.4. Spatial Autocorrelation between Crashes in Different Conditions

The global Moran's *I* method was employed to determine the global spatial autocorrelations of crashes for both directions. As highlighted in Table 3, there is spatial autocorrelation of crashes in the Qazvin-to-Abyek direction, with more than 95% confidence, and the crashes are clustered. However, there is a random form for the opposite direction. The global Moran's *I* autocorrelation method could also be conducted for different conditions, including spatiotemporal conditions, and test the effects of some parameters—namely, hourly traffic volume and traffic speed—on the SDC. This method would show global spatial autocorrelation for different traffic volumes, speed ranges, and times of the day. According to the results provided in Table 4, there are spatial clusters in some conditions. For instance, in the Abyek-to-Qazvin direction, there is a spatial crash clustering only in the high range of hourly traffic volume, whereas there is a random spatial autocorrelation of crashes for other conditions.

In the Qazvin-to-Abyek direction, there is a spatial crash clustering with 95% confidence for the low range of hourly traffic volume, high range of speed, and time from 03 to 06, while for other conditions the crashes occur randomly. The location of the crash clusters for all conditions was the same and close to 52 km. To further investigate the causes of these conditions, the geometric characteristics of this path—such as curves, slopes, cross-sections, and vertical alignment—were reviewed. The results indicated that the only specific observed case in that segment was a high downhill (3%) slope.

**Table 3.** The global Moran's *I* for spatial crash frequency.

| Direction | Moran's *I* | Z | *p*-Value | Spatial Distribution (95% Confidence) |
|---|---|---|---|---|
| Qazvin to Abyek | 0.436746 | 2.553551 | 0.010663 | *Clustered* |
| Abyek to Qazvin | 0.155543 | 0.991892 | 0.32125 | Random |

**Table 4.** The Global Moran's *I* for spatial crash frequency in different conditions.

| | | Condition | Moran's *I* | Z | *p*-Value | Spatial Distribution (95% Confidence) |
|---|---|---|---|---|---|---|
| Direction | Qazvin to Abyek | V < 1107 | 0.489669 | 2.846972 | 0.004414 | *Clustered* |
| | | 1107 < V < 1982 | 0.153117 | 0.912806 | 0.361345 | Random |
| | | V > 1982 | 0.124857 | 0.823357 | 0.410305 | Random |
| | | S < 97 | 0.074382 | 0.527389 | 0.597924 | Random |
| | | S > 97 | 0.698778 | 3.989106 | 0.000066 | *Clustered* |
| | | 00–03 | 0.066896 | 0.492272 | 0.622527 | Random |
| | | 03–06 | 0.524641 | 3.042979 | 0.002342 | *Clustered* |
| | | 06–09 | 0.177791 | 1.094382 | 0.273788 | Random |
| | | 09–12 | 0.045235 | 0.352046 | 0.724804 | Random |
| | | 12–15 | −0.203112 | −1.036415 | 0.300009 | Random |
| | | 15–18 | 0.181377 | 1.109924 | 0.267032 | Random |
| | | 18–21 | 0.089734 | 0.621703 | 0.534137 | Random |
| | | 21–24 | 0.003544 | 0.12394 | 0.901363 | Random |
| | Abyek to Qazvin | V < 1378 | 0.102881 | 0.703368 | 0.481827 | Random |
| | | 1378 < V < 1982 | 0.055929 | 0.417969 | 0.67597 | Random |
| | | V > 2067 | 0.393178 | 2.33552 | 0.019516 | *Clustered* |
| | | S < 97 | 0.202809 | 1.347552 | 0.177802 | Random |
| | | S > 97 | −0.114638 | −0.5438 | 0.586579 | Random |
| | | 00–03 | 0.153025 | 0.985104 | 0.324573 | Random |
| | | 03–06 | −0.141334 | −0.700012 | 0.48392 | Random |
| | | 06–09 | −0.010544 | 0.044112 | 0.964815 | Random |
| | | 09–12 | −0.045932 | −0.159563 | 0.873225 | Random |
| | | 12–15 | −0.151469 | −0.75118 | 0.452544 | Random |
| | | 15–18 | 0.063173 | 0.464509 | 0.642283 | Random |
| | | 18–21 | 0.160602 | 0.778682 | 0.436167 | Random |
| | | 21–24 | 0.057169 | 0.443945 | 0.657082 | Random |

Note: In the Qazvin to Abyek section, the condition groups are Volume (V) (veh/h), Speed (S) (km/h), and Time (hours). In the Abyek to Qazvin section, the condition groups are Volume (V) (veh/h), Speed (S) (km/h), and Time (hours).

According to Iran's Highway Geometric Design Code, this slope, with a length of about 5.5 km, is located at the upper limit of the allowable slope. The long length and high percentage of the downhill slope are influential factors in increasing the number of heavy vehicle crashes [43]. Therefore, in the next step, the effect of the percentage of heavy vehicles on the SDC was investigated. The average percentage of heavy vehicles was 16% for the whole condition. To examine the impact of heavy vehicles, the SDC was analyzed in two modes: the lower percentage (>16%) and the higher percentage than the average of heavy vehicles (>16%). The output of the analysis is shown in Figure 11. The results showed that when the percentage of heavy vehicles was high, the highest crash frequency occurred around the 52 km segment; meanwhile, in the case of the low percentage of heavy vehicles, some other parts had a greater crash frequency. Then, the effect of the heavy vehicles in crashes (responsible and non-culpable) was analyzed (Table 5). The investigation of the effect of the percentage of heavy vehicles involved in crashes indicated that, in general, the percentage of heavy vehicles responsible for crashes was approximately 13.7%. The percentage of non-culpable heavy vehicles involved in accidents was about

18.5%. However, these values, which differed from the average values, were equal to 18.4% and 25% between 03 and 06 in the morning. Thus, the effect of the heavy vehicle percentage on the clustering of accidents in the above conditions was confirmed.

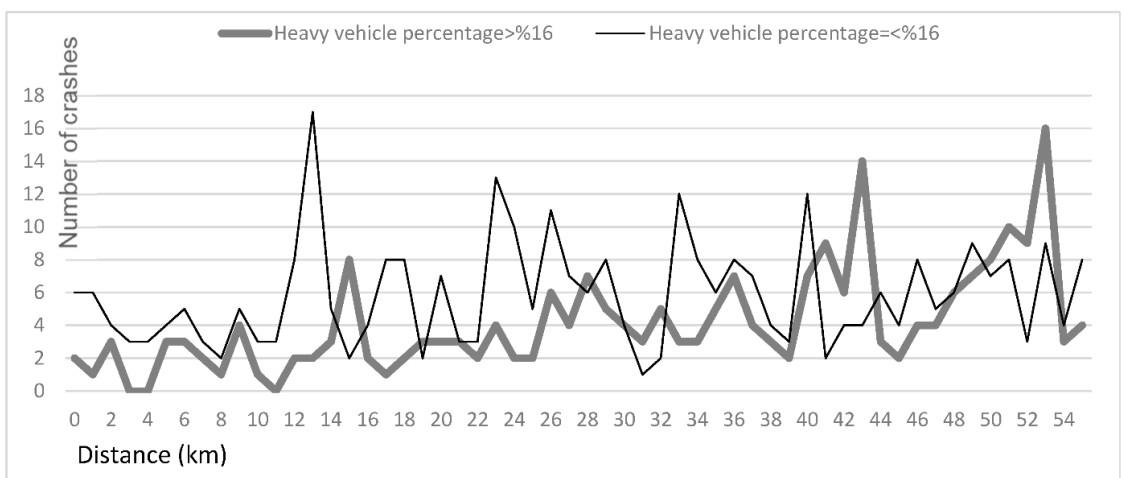

**Figure 11.** Impact of heavy vehicle percentage on the SDC.

**Table 5.** Crash number and percentage details for heavy vehicles.

| Crash Percentage for Heavy Vehicle Non-Culpable | Crash Number for Heavy Vehicle Non-Culpable | Crash Percentage for Heavy Vehicle Responsible | Crash Number for Heavy Vehicle Responsible | Crash Frequency | Time |
|---|---|---|---|---|---|
| 0.185507246 | 128 | 0.137681159 | 95 | 690 | 0–24 |
| 0.25 | 19 | 0.184210526 | 14 | 76 | 03–06 |

Several techniques are available to investigate the correlation between these different parameters. Among them, Pearson's correlation test is one of the simplest and most well-known methods. Hence, this study used Pearson's correlation test technique. It is noteworthy that the correlation coefficients near 1 denote high correlations, while those parameters with correlation coefficients near 0 are non-correlated. According to the results shown in Table 6, there is very strong correlation between low traffic volume and high heavy vehicle percentage, as well as between low traffic volume and high traffic mean speed (Pearson's correlation > 0.8). There is a strong correlation between high heavy vehicle percentage and high traffic mean speed (0.8 > Pearson's correlation > 0.6). All correlations between parameters were significant with 95% confidence. The results indicated that these parameters had similar crash patterns for parts, i.e., the segments had almost the same crash conditions for high average traffic speed, low hourly traffic volume, and high heavy vehicle percentage. The time 03–06 had a medium correlation with the other parameters. Since this result was not just for hotspot segments but for whole parts of the road, it can be concluded that these parameters have a practical simultaneous effect on the number of crashes.

**Table 6.** Pearson's correlation for different conditions.

| | LV | HS | T0306 | HVP |
|---|---|---|---|---|
| LV | 1 | 0.801 ** | 0.568 ** | 0.891 ** |
| HS | 0.801 ** | 1 | 0.487 ** | 0.701 ** |
| T0306 | 0.568 ** | 0.487 ** | 1 | 0.565 ** |
| HVP | 0.891 ** | 0.701 ** | 0.565 ** | 1 |

**. Correlation is significant at the 0.01 level (2-tailed). LV: low hourly traffic volume (veh/h); HS: high mean traffic speed (km/h). T0306: time 03–06; HVP: heavy vehicle percentage > 16%.

## 5. Conclusions

The main objective of this study was to investigate a method for a freeway in Iran to help traffic managers identify the high-risk times, locations, and conditions to make proper decisions. The primary step of this study was to conduct spatial, spatiotemporal, and density analyses of the crashes. Various methods, including PKDE, local Moran's *I*, and Getis-Ord *Gi*\*, were employed to identify the hotspots and high-crash-density segments along the route. The results revealed different hotspots and high clusters for different times. Then, the effects of some traffic parameters—namely, hourly traffic volume and speed—on the SDC were investigated globally by global Moran's *I* and locally by the PKDE, local Moran's *I*, and Getis-Ord *Gi*\* methods. The findings revealed different hotspots and high clusters for different traffic conditions, and some sections were found to be more likely to experience a crash.

Moreover, according to the findings, the combination of high mean traffic speed, low traffic volume, and high downhill slope in the early morning could increase the odds of crashes, especially for heavy vehicles. These conditions caused crash clustering. These results could help traffic managers to make a suitable decision when these conditions occur. Such management could be developed into real-time traffic management systems that predict real-time crashes and estimate the crash risk using high-resolution detector data [44,45]. For these conditions, further attention is required, namely, specific speed limitations, more police control, prohibition of the entry of heavy vehicles on special days and situations, and increasing drivers' awareness of high-risk states through VMS or other on-trip information systems. Such a real-time traffic management system could be developed in future studies.

This study focused only on fatality- and injury-related crash data because the damage-only crash data were not accessible, which was one important limitation of this research. Access to the above information could increase the accuracy of future studies.

In addition, further studies investigating the impact rates of different factors via modeling could help us to learn more about the factors affecting the clustering of crashes. Modeling based on investigating the role of geometric road characteristics could also be beneficial. Determining the extent to which these factors affect STDC could be the focus of future studies. The influence of factors other than SDC and STDC, including human characteristics and weather conditions, could be further examined. Although the present study could be extended in many ways, the analyses developed in this paper are applicable to other similar freeways and provide useful information for traffic management.

**Author Contributions:** Methodology, K.Z. and A.O.; Formal analysis, K.Z.; Validation, A.T.K.; Writing—review & editing, K.Z. and A.O.; Project administration, A.T.K. All authors have read and agreed to the published version of the manuscript.

**Funding:** No funding was received for conducting this study, and the authors have no relevant financial or non-financial interests to disclose.

**Data Availability Statement:** The data for this study are not available due to legal restrictions.

**Conflicts of Interest:** The authors declare that there are no conflicts of interest regarding the publication of this paper.

## Appendix A

Local Moran's *I* output for SDC crashes:

**Table A1.** Local Moran's I output for Qazvin-to-Abyek SDC crashes.

| SOURCE_ID | Accident | | | | Low Volume | | | | Mid Volume | | | | High Volume | | | | Low Speed | | | | High Speed | | | |
|---|---|---|---|---|---|---|---|---|---|---|---|---|---|---|---|---|---|---|---|---|---|---|---|---|
| | LMiIndex | LMiZScore | LMiPValue | COType | LMiIndex | LMiZScore | LMiPValue | COType | LMiIndex | LMiZScore | LMiPValue | COType | LMiIndex | LMiZScore | LMiPValue | COType | LMiIndex | LMiZScore | LMiPValue | COType | LMiIndex | LMiZScore | LMiPValue | COType |
| 7 | 0.0076 | 2.04 | 0.0417 | LL | 0.0039 | 1.07 | 0.2848 | | 0.0027 | 0.74 | 0.4610 | | 0.0010 | 0.29 | 0.7746 | | 0.0014 | 0.40 | 0.6873 | | 0.0036 | 0.97 | 0.3324 | |
| 8 | 0.0078 | 2.10 | 0.0353 | LL | 0.0042 | 1.15 | 0.2508 | | 0.0028 | 0.76 | 0.4467 | | 0.0010 | 0.29 | 0.7747 | | 0.0014 | 0.41 | 0.6843 | | 0.0033 | 0.89 | 0.3739 | |
| 12 | −0.0007 | −0.68 | 0.4943 | | 0.0005 | 0.49 | 0.6220 | | 0.0002 | 0.21 | 0.8352 | | 0.0063 | 6.52 | 0.0000 | HH | 0.0031 | 3.17 | 0.0015 | HH | 0.0007 | 0.72 | 0.4725 | |
| 13 | −0.0018 | −1.29 | 0.1982 | | 0.0008 | 0.60 | 0.5472 | | 0.0001 | 0.07 | 0.9453 | | 0.0045 | 3.35 | 0.0008 | HH | 0.0061 | 4.52 | 0.0000 | HH | 0.0016 | 1.14 | 0.2525 | |
| 14 | −0.0011 | −1.12 | 0.2627 | | 0.0003 | 0.35 | 0.7259 | | −0.0001 | −0.11 | 0.9108 | | −0.0018 | −1.83 | 0.0676 | | 0.0031 | 3.17 | 0.0015 | HH | 0.0009 | 0.89 | 0.3761 | |
| 23 | 0.0004 | 0.45 | 0.6499 | | 0.0000 | 0.00 | 0.9963 | | −0.0005 | −0.47 | 0.6368 | | 0.0036 | 3.83 | 0.0001 | HH | 0.0019 | 2.02 | 0.0431 | HH | 0.0000 | 0.01 | 0.9916 | |
| 24 | 0.0000 | 0.06 | 0.9558 | | 0.0001 | 0.07 | 0.9451 | | −0.0004 | −0.24 | 0.8072 | | 0.0034 | 2.55 | 0.0108 | HH | 0.0017 | 1.24 | 0.2141 | | −0.0001 | −0.04 | 0.9644 | |
| 27 | −0.0001 | −0.08 | 0.9372 | | 0.0000 | 0.03 | 0.9757 | | 0.0020 | 2.05 | 0.0405 | HH | −0.0001 | −0.06 | 0.9527 | | 0.0011 | 1.16 | 0.2461 | | 0.0000 | −0.03 | 0.9752 | |
| 30 | −0.0001 | −0.01 | 0.9915 | | −0.0001 | −0.06 | 0.9536 | | 0.0000 | 0.01 | 0.9909 | | 0.0005 | 0.42 | 0.6724 | | 0.0026 | 1.97 | 0.0491 | LL | 0.0003 | 0.27 | 0.7890 | |
| 33 | 0.0005 | 0.40 | 0.6881 | | −0.0003 | −0.19 | 0.8524 | | 0.0028 | 2.05 | 0.0406 | HH | 0.0001 | 0.12 | 0.9073 | | −0.0002 | −0.11 | 0.9126 | | −0.0001 | −0.06 | 0.9549 | |
| 34 | 0.0013 | 1.01 | 0.3143 | | −0.0008 | −0.57 | 0.5693 | | 0.0036 | 2.61 | 0.0089 | HH | −0.0002 | −0.09 | 0.9308 | | −0.0001 | −0.04 | 0.9684 | | 0.0006 | 0.47 | 0.6407 | |
| 41 | −0.0002 | −0.14 | 0.8900 | | 0.0044 | 3.22 | 0.0013 | HH | −0.0007 | −0.46 | 0.6471 | | 0.0003 | 0.26 | 0.7986 | | 0.0000 | 0.03 | 0.9751 | | 0.0008 | 0.60 | 0.5476 | |
| 49 | 0.0027 | 1.99 | 0.0462 | HH | 0.0045 | 3.29 | 0.0010 | HH | −0.0010 | −0.69 | 0.4907 | | 0.0001 | 0.12 | 0.9044 | | −0.0008 | −0.57 | 0.5659 | | 0.0020 | 1.48 | 0.1395 | |
| 50 | 0.0019 | 1.94 | 0.0529 | | 0.0033 | 3.37 | 0.0008 | HH | −0.0014 | −1.37 | 0.1699 | | 0.0002 | 0.27 | 0.7849 | | 0.0003 | 0.31 | 0.7528 | | 0.0031 | 3.13 | 0.0017 | HH |
| 51 | 0.0022 | 2.22 | 0.0262 | HH | 0.0022 | 2.22 | 0.0264 | HH | 0.0001 | 0.11 | 0.9140 | | −0.0005 | −0.54 | 0.5887 | | −0.0001 | −0.08 | 0.9387 | | −0.0001 | −0.09 | 0.9255 | |
| 52 | 0.0050 | 3.67 | 0.0002 | HH | 0.0043 | 3.17 | 0.0015 | HH | 0.0007 | 0.55 | 0.5823 | | −0.0033 | −2.43 | 0.0152 | LH | 0.0008 | 0.62 | 0.5325 | | −0.0002 | −0.13 | 0.8938 | |
| 53 | 0.0000 | 0.04 | 0.9702 | | 0.0006 | 0.35 | 0.7228 | | −0.0014 | −0.77 | 0.4412 | | −0.0051 | −2.99 | 0.0028 | HL | −0.0046 | −2.67 | 0.0077 | HL | 0.0087 | 5.04 | 0.0000 | HH |
| 54 | −0.0028 | −1.93 | 0.0530 | | −0.0016 | −1.07 | 0.2828 | | −0.0020 | −1.37 | 0.1699 | | −0.0023 | −1.64 | 0.1016 | | −0.0055 | −3.84 | 0.0001 | LH | 0.0088 | 6.09 | 0.0000 | HH |

**Table A2.** Local Moran's I output for Abyek-to-Qazvin SDC crashes.

| SOURCE_ID | Accident | | | | Low Volume | | | | Mid Volume | | | | High Volume | | | | Low Speed | | | | High Speed | | | |
|---|---|---|---|---|---|---|---|---|---|---|---|---|---|---|---|---|---|---|---|---|---|---|---|---|
| | LMiIndex | LMiZScore | LMiPValue | COType | LMiIndex | LMiZScore | LMiPValue | COType | LMiIndex | LMiZScore | LMiPValue | COType | LMiIndex | LMiZScore | LMiPValue | COType | LMiIndex | LMiZScore | LMiPValue | COType | LMiIndex | LMiZScore | LMiPValue | COType |
| 4 | −0.0010 | −1.03 | 0.3026 | | −0.0010 | −1.04 | 0.2975 | | 0.0005 | 0.56 | 0.5784 | | 0.0001 | 0.13 | 0.8990 | | 0.0004 | 0.49 | 0.6240 | | −0.0022 | −2.20 | 0.0275 | LH |
| 5 | 0.0016 | 1.19 | 0.2335 | | 0.0031 | 2.33 | 0.0196 | HH | 0.0045 | 3.25 | 0.0012 | HH | −0.0002 | −0.10 | 0.9205 | | 0.0043 | 3.38 | 0.0007 | HH | −0.0013 | −0.89 | 0.3711 | |
| 6 | −0.0014 | −0.80 | 0.4258 | | −0.0036 | −2.10 | 0.0357 | HL | 0.0009 | 0.53 | 0.5927 | | −0.0009 | −0.47 | 0.6380 | | 0.0004 | 0.28 | 0.7780 | | −0.0009 | −0.50 | 0.6145 | |
| 13 | 0.0003 | 0.35 | 0.7298 | | 0.0003 | 0.30 | 0.7656 | | 0.0010 | 1.03 | 0.3029 | | −0.0005 | −0.45 | 0.6521 | | 0.0023 | 2.49 | 0.0126 | HH | −0.0005 | −0.44 | 0.6583 | |
| 17 | −0.0003 | −0.16 | 0.8729 | | 0.0031 | 2.33 | 0.0197 | HH | 0.0000 | 0.01 | 0.9918 | | 0.0008 | 0.59 | 0.5538 | | 0.0011 | 0.89 | 0.3743 | | 0.0003 | 0.21 | 0.8365 | |
| 24 | 0.0016 | 1.21 | 0.2269 | | 0.0039 | 2.97 | 0.0030 | HH | −0.0011 | −0.76 | 0.4489 | | −0.0001 | −0.04 | 0.9651 | | 0.0002 | 0.19 | 0.8500 | | 0.0012 | 0.92 | 0.3558 | |
| 32 | 0.0000 | 0.01 | 0.9887 | | 0.0003 | 0.34 | 0.7333 | | 0.0022 | 2.28 | 0.0229 | HH | −0.0001 | −0.11 | 0.9128 | | 0.0000 | 0.00 | 0.9988 | | 0.0009 | 0.95 | 0.3408 | |
| 36 | 0.0005 | 0.41 | 0.6801 | | 0.0008 | 0.65 | 0.5131 | | 0.0015 | 1.13 | 0.2578 | | −0.0004 | −0.23 | 0.8162 | | 0.0026 | 2.04 | 0.0412 | LL | −0.0026 | −1.86 | 0.0625 | |
| 39 | 0.0000 | −0.03 | 0.9762 | | −0.0004 | −0.36 | 0.7179 | | 0.0004 | 0.41 | 0.6836 | | 0.0009 | 0.91 | 0.3603 | | 0.0000 | 0.03 | 0.9793 | | 0.0024 | 2.45 | 0.0142 | |
| 43 | −0.0006 | −0.42 | 0.6779 | | −0.0010 | −0.71 | 0.4767 | | −0.0030 | −2.12 | 0.0340 | HL | 0.0001 | 0.13 | 0.8969 | | −0.0004 | −0.27 | 0.7845 | | −0.0040 | −2.83 | 0.0046 | HL |
| 44 | −0.0008 | −0.85 | 0.3961 | | −0.0006 | −0.64 | 0.5236 | | −0.0026 | −2.59 | 0.0097 | LH | −0.0007 | −0.74 | 0.4594 | | −0.0007 | −0.71 | 0.4795 | | −0.0020 | −1.98 | 0.0473 | LH |
| 49 | 0.0008 | 0.61 | 0.5402 | | 0.0011 | 0.82 | 0.4141 | | −0.0008 | −0.58 | 0.5641 | | 0.0060 | 4.38 | 0.0000 | HH | 0.0002 | 0.16 | 0.8706 | | 0.0010 | 0.72 | 0.4721 | |
| 50 | 0.0033 | 2.46 | 0.0140 | HH | 0.0000 | 0.06 | 0.9501 | | −0.0035 | −2.49 | 0.0126 | LH | 0.0059 | 4.28 | 0.0000 | HH | 0.0007 | 0.58 | 0.5623 | | 0.0016 | 1.14 | 0.2527 | |
| 51 | 0.0048 | 3.55 | 0.0004 | HH | −0.0005 | −0.32 | 0.7502 | | −0.0010 | −0.73 | 0.4660 | | −0.0008 | −0.55 | 0.5827 | | 0.0000 | 0.00 | 0.9985 | | 0.0018 | 1.36 | 0.1738 | |
| 54 | 0.0042 | 2.65 | 0.0081 | LL | 0.0027 | 1.72 | 0.0855 | | 0.0007 | 0.46 | 0.6442 | | 0.0024 | 1.50 | 0.1342 | | 0.0026 | 1.74 | 0.0819 | | 0.0018 | 1.13 | 0.2585 | |
| 55 | 0.0041 | 3.17 | 0.0015 | LL | 0.0025 | 1.96 | 0.0503 | | 0.0006 | 0.47 | 0.6406 | | 0.0022 | 1.70 | 0.0886 | | 0.0017 | 1.37 | 0.1707 | | 0.0030 | 2.28 | 0.0224 | LL |

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
