# Peer review of "Influence of Traffic Parameters on the Spatial Distribution of Crashes on a Freeway to Increase Safety"

_sustainability, doi:10.3390/su15010493_

Round 1
Reviewer 1 Report
This paper investigated a method to help traffic managers to identify high-risk times, locations, and conditions to make proper decisions, Various methods were employed and compared to conduct the crash analyses, and the findings an results are helpful for the traffic management and future further research, it is good enough to be published in current condition.
Author Response
We appreciate the reviewer for his/her positive comments.
Reviewer 2 Report
It is recommended that the manuscript be formatted following the journal template. The manuscript has some grammatical errors and typos. All acronyms should be written in expanded form when they are first included in the text .
How do pedestrians affect this problem? and the surrounding environment or socio-cultural aspects?
we recommend reading the following works
1) Erdogan, S. (2009). Explorative spatial analysis of traffic accident statistics and road mortality among the provinces of Turkey. Journal of safety research, 40(5), 341-351.
2) Kashani, A. T., & Zandi, K. (2020). Influence of traffic parameters on the temporal distribution of crashes. KSCE Journal of Civil Engineering, 24(3), 954-961.
3)Deluka-Tibljaš, A., Ištoka Otković, I., Campisi, T., & Šurdonja, S. (2021). Comparative analyses of parameters influencing children pedestrian behavior in conflict zones of urban intersections. Safety, 7(1), 5.
The novelty of the research should be more emphasized in the introduction of the manuscript.
Figures 2 through 7 should be followed by more comments, and axes should be corrected by inserting the specifics of the units of measurement represented
More text should be inserted in the introductory part by inserting more bibliographical references specifying the topic covered .
In the concluding part the limitations of the research and possible future steps should be inserted .
The manuscript does not emphasize the case studies examined therefore more description of the locations and temporality afferent to the case study
Author Response
We appreciate the reviewer for his/her valuable comments that helped us to revise the paper. We have made revisions to improve the quality of our manuscript. Each comment from the reviewer is now presented as attached file, followed by the point-by-point responses and actions taken in the revision. Changes in the manuscript are identified by using the track changes mode in MS Word and colored text with a comment note attached to it.

Reviewer 3 Report
the manuscript investigates distribution of crash data based on spatial analyses. the work is interesting but there are several issues that should be considered as follows:
1.the research gap in this field and the novelty and contributions were not specified
2. the spatial analysis and statistics covers an extensive area, so it is not appropriate to say that we select some methods in this area and investigate them. Authors should check which part of spatial analysis they need and choose the best methods in that part with proper justification. for example you select point pattern analysis and investigate them by kernel density estimation
3. it is better to present some quantitative results in the abstract
4. it is better to present the case study and data after presenting the methods and methodology. the data can be explained in more detailed. for example are crash data gathered in point data or segment data?
5. the manuscript should present a clear methodology in a specific section or sub-section. the workflow in the present manuscript is missed
6. why did the authors use the PKDE method instead of NKDE one since the data are network data?
7. each method used in the manuscript should be justified. for example why did the authors use the autocorrelation methods
8. the parameters of each method should be clarified. for example how did the authors adjusted the kernel function or the weight matrix
9. the local Morans' I was developed for spatio-temporal analysis. why the author did not use that index?
10. why did the author classified the traffic speed in 2 classes and not other number? it should be explained. what is the classification method?
11. it is better to discuss the results in more detailed. for example the parameters in the line 374 can be investigated in the the crash data
12. the percentages presented in lines 388-390 are based on your data or other studies. it should be specified
Author Response

(The authors gave the same response as above.)

Round 2
Reviewer 2 Report
No other comments.The paper is elegible for pubblication
Author Response
Response to Reviewer #2 (round 2)
We appreciate the reviewer for his/her valuable comments that helped us to revise the paper. We have made revisions to improve the “English language and style”. Changes in the manuscript are identified by using the track changes mode in MS Word and colored text with a comment note attached to it.
Reviewer 3 Report
The comments are answered appropriately but there are some comments as follows
1. The novelty should be revised. The manuscript investigated the point pattern of crash data and then related that result with some parameters. In other words the manuscript did not analyze the speed and traffic data with spatial analysis
2. The results of NKDE will be presented in revised manuscript although the results are the same as PKDE
3. There are three subsets data in the manuscript but in the aswer file there are two subsets
4. One paragraph should be given in the manuscript for describing the flowchart of the research
Author Response
Response to Reviewer #3 (round 2)
The comments are answered appropriately but there are some comments as follows
We appreciate the reviewer for his/her valuable time to review our manuscript again. We have made revisions to improve the quality of our manuscript. Each comment from the reviewer is now presented below, followed by the point-by-point responses and actions taken in the revision. Changes in the manuscript are identified by using the track changes mode in MS Word and colored text with a comment note attached to it, describing what is changed and why. Also, a cover letter describing how we have specifically addressed each of the reviewer comments is supplied with our revision.
Comment 3-1 R2
1. The novelty should be revised. The manuscript investigated the point pattern of crash data and then related that result with some parameters. In other words the manuscript did not analyze the speed and traffic data with spatial analysis
Thank you for your attention. Actually, the data are local information at the point of the crash. But, at the last stages of the analysis, we related the data to a specific origin (which was the first point on the road). In other words, we converted the point crash data to spatial ones. As the text was vague, based on your valuable comment we changed the text as follows:
“It should be noted that in the analyses, the point traffic information at the crash location is employed. But, since at the last stage of the procedure they are related to a specific origin (and not separate road segments) they can be considered as spatial information.”
Comment 3-2 R2
- The results of NKDE will be presented in revised manuscript although the results are the same as PKDE
A very good point has been mentioned in this comment, we added NKDE analysis too.
Comment 3-3 R2
- There are three subsets data in the manuscript but in the aswer file there are two subsets
We appreciate this thoughtful comment. We corrected the text.
Comment 3-4 R2
- One paragraph should be given in the manuscript for describing the flowchart of the research
Thanks for your very good comment. We added some parts to the flowchart, such as:
“Impact of traffic parameters on SDC (such as volume and speed)"
And also, more explanation about the flowchart is added as below:
“The methodology of the present study is shown as a flowchart in Figure 1. In the rest of this section, the different parts of the methodology have been explained. It is worth mentioning that steps 3 and 4 have been previously performed by the authors and can be followed in Kashani and Zandi [25], the rest of the steps were reported in this study. Other steps of Figure 1 were the main focus of this research, which are given in the following. Step 1 is obtaining data which is explaining about the two data sets, and how to achieve and step 2 is the merge of these two data series. More explanations about steps 1 and 2 are located in section 3 (Case Study and Data). Step 5 is about spatial analysis and step 6 is about the testing significance and their methods, which both steps are given in sections 2.1 (Spatial Analysis) and 2.2 (High-Frequency Crash Location). Step 7 investigates the Spatio-temporal analysis which is explained the correlation of time and location of the crashes which are explained in section 1 (Introduction). In the rest of this section, the different parts of the methodology have been explained."